# A Narrative Role of Vitamin D and Its Receptor: With Current Evidence on the Gastric Tissues

**DOI:** 10.3390/ijms20153832

**Published:** 2019-08-05

**Authors:** Shaima Sirajudeen, Iltaf Shah, Asma Al Menhali

**Affiliations:** 1Department of Biology, College of Science, United Arab Emirates University (UAEU), Al Ain P.O. Box 15551, UAE; 2Department of Chemistry, College of Science, United Arab Emirates University (UAEU), Al Ain P.O. Box 15551, UAE

**Keywords:** 1α,25(OH)_2_D, vitamin D deficiency, vitamin D epimers, cytochrome P450, 1,25-MARRS, stomach

## Abstract

Vitamin D is a major steroid hormone that is gaining attention as a therapeutic molecule. Due to the general awareness of its importance for the overall well-being, vitamin D deficiency (VDD) is now recognized as a major health issue. The main reason for VDD is minimal exposure to sunlight. The vitamin D receptor (VDR) is a member of the steroid hormone receptors that induces a cascade of cell signaling to maintain healthy Ca^2+^ levels that serve to regulate several biological functions. However, the roles of vitamin D and its metabolism in maintaining gastric homeostasis have not yet been completely elucidated. Currently, there is a need to increase the vitamin D status in individuals worldwide as it has been shown to improve musculoskeletal health and reduce the risk of chronic illnesses, including some cancers, autoimmune and infectious diseases, type 2 diabetes mellitus, neurocognitive disorders, and general mortality. The role of vitamin D in gastric homeostasis is crucial and unexplored. This review attempts to elucidate the central role of vitamin D in preserving and maintaining the overall health and homeostasis of the stomach tissue.

## 1. Introduction

Vitamin D deficiency (VDD) is a global health issue. More than 1 billion people suffer from vitamin D deficiency or insufficiency [1]. *Endocrine Connections* reported a definite increase in the number of individuals affected with VDD in recent decades [2]. The susceptible risk groups include people who spend more time indoors [3], like the elderly and those who are homebound [4], people with pigmented skins, vegans [5], and pregnant women and their developing children [6,7].

The regions most affected by this disease comprise China, India, Mongolia, and the middle eastern countries including the UAE, Saudi Arabia, Jordan, and Qatar [8]. Despite receiving plentiful sunshine, Vitamin D status is markedly low in the Middle East, for example, over 90% of the UAE population is affected [9]. The deficiency is more pronounced in women than in men in the Middle East, chiefly due to their traditional clothing style of covering the skin completely [10]. The intense heat also contributes to VDD in people staying indoors and in excessive users of sunscreens [11].

The discovery of Vitamin D and its role in skeletal health can be traced back to the middle of the 17th century, to the English physician, Francis Glisson. He published a study titled “A treatise of rickets, as a disease common to children”, wherein he described rickets as an “absolutely new disease that has never been described in any of the ancient or modern books” [12]. The role of diet in this disease was not known at that time [13]. Several years later, Sir Edward Mellanby (1921) ascribed the cause of rickets to the deficiency of a fat-soluble vitamin [14]. However, the breakthrough discovery was when McCollum and his associates found that this fat-soluble entity was a dietary supplement and named it Vitamin D [15].

The detection and role of the vitamin D receptor (VDR) were followed thereafter in various human and animal tissues. This led to determining the actual role of vitamin D in health and disease. This query led to understanding the role of vitamin D in the development, metabolism, and well-being of the organism, particularly in calcium homeostasis and bone maintenance [16,17]. The non-classical functions of vitamin D and its importance in human physiology became evident over the past few decades [16,17,18]. Vitamin D is synthesized by cells of the immune system and controls the immune function along with anti-bacterial activities [18,19]. It also plays a role in anti-proliferative and anti-tumorigenic activities in the cancer cells of the breast, colon, skin, stomach, and prostate [20,21]. Low levels of vitamin D have been coupled to various epidemiological states such as inflammatory bowel disease and autoimmune disorders including multiple sclerosis, inflammation [22], chronic kidney disease [23], various types of cancer [20], cardiovascular disease [24], obesity [25], mortality [26], type 1 diabetes mellitus [27], and hypertension [28].

Vitamin D status is assessed by determining the amount of total 25-hydroxyvitamin D (25(OH)D) in circulation. The adequate levels of total 25(OH)D are between 50 and 70 ng/mL. In VDD, the range is less than 30 ng/mL (Table 1) [27,29].

There have been a few methods used for the clinical detection of total 25(OH)D. Most of these methods measure total 25(OH)D along with other hydroxylated vitamin D metabolites in human serum. Vitamin D metabolites in circulation at a given time account for 2‒20% of the total 25(OH)D present. Therefore, the active levels of vitamin D could be misinterpreted.

The common detection methods of vitamin D in clinical laboratories include high-performance liquid chromatography (HPLC), radioimmunoassay (RIA), enzyme-linked immunoassay (ELISA), and quantitative chemiluminescent immunoassay (CLIA). A kit-based method of RIA developed by DiaSorin S.p.A (Saluggia, Italy) is the gold standard for total 25(OH)D measurements and is used to establish the reference range of serum vitamin D [30,31]. Lately, liquid chromatography-tandem mass spectrometry (LC-MS/MS) is gaining attention due to its application in measuring the various forms of vitamin D and the D3 epimers in the serum of people of all ages [32]. This method is labor-intensive and technically strenuous, but is earning extensive credence. Many publications evaluating the different methods to determine serum vitamin D levels authenticate the fact that this method is cost-effective and highly precise [30,33].

People are given vitamin D supplements to overcome VDD for specific periods, depending on the nature and severity of the scarcity [34]. Global awareness of the health benefits of vitamin D are alarmingly low. For example, there are a few studies investigating the role of vitamin D or its receptor in the gastrointestinal tract (GIT). This review discusses the general characteristics, function, and importance of vitamin D, its metabolism and the two different forms of the VDRs. The second half of the review highlights the role of vitamin D in the stomach tissue and its function in maintaining the homeostasis of the gastric environment and the preservation of gastric health.

## 2. Vitamin D Ligands

Vitamin D is a hormone. Its active form is commonly known as 1α,25-dihydroxyvitamin D (1α,25(OH)_2_D). The latest research reveals a dual role for 1α,25(OH)_2_D, which is generated as an activator or inhibitor, depending on the location and function of the target cells [35]. Vitamin D is a fat-soluble vitamin existing mainly in two forms, ergocalciferol (Vitamin D2, 1α,25(OH)_2_D2), which is of plant origin, and cholecalciferol, (Vitamin D3, 1α,25(OH)_2_D3), which is of animal origin (Figure 1) [36]. The presence of 1α,25(OH)_2_D2 has been confirmed in several members of the plant kingdom, including algae, fungi, and in plants contaminated with certain types of fungi. 1α,25(OH)_2_D3 on the other hand, is found mostly in dairy products, fatty fish, beef liver, and egg yolks [35].

Askew et al. identified the structure of 1α,25(OH)_2_D2 in 1931. Later, the structure of 1α,25(OH)_2_D3 was determined through synthetic means [37]. The findings from the second half of the 20th century showed that vitamin D is a prohormone [38]. Prohormones are inactive precursors of hormones that are converted to active anabolic forms with the help of enzymatic activities in the body [39]. Vitamin D is photochemically produced in the skin from 7-dehydrocholesterol [40]. Subsequent chemical reactions in the liver and kidneys convert 7-dehydrocholesterol into the active form that exerts biological functions by binding to the VDRs.

Maestro et al., (2016) reported that VDR is found in most human tissues and has more than 1000 target genes [41]. 1α,25(OH)_2_D exerts its biological action by binding to VDR and generates biological effects through genomic and rapid non-genomic responses. The details of the two types of VDRs are discussed in Section 3.1 and Section 3.2.

### 2.1. Vitamin D Structure and Biosynthesis

1α,25(OH)_2_D is a secosteroid with a broken 9,10 carbon-carbon bond in the B ring of the cyclopentanoperhydrophenanthrene structure, which is a carbon skeleton found in steroids. The main structural difference between 1α,25(OH)_2_D2 and 1α,25(OH)_2_D3 is the presence of a double bond and a methyl group on the side chain of vitamin D2 (Figure 1) [20,42]. Norman et al. (1982) reported that the structurally flexible nature of 1α,25(OH)_2_D molecule helps it to successfully fit into the ligand binding domains (LBD) of the vitamin D binding protein (DBP), nuclear VDR (nVDR), and membrane VDR (mVDR), mainly depending on the function of 1α,25(OH)_2_D [43]. 1α,25(OH)_2_D can exist in both bowl-like and planar-like shapes, which support a mechanism by which it can induce many cellular responses [44].

1α,25(OH)_2_D3 is synthesized through the action of sunlight along with subsequent enzymatic reactions. 1α,25(OH)_2_D2 is formed by ultraviolet B (UVB) irradiation of the ergosterol in plants and fungi (Figure 2).

The skin synthesizes 1α,25(OH)_2_D3 from 7-dehydrocholesterol, by the photochemical conversion of pre-vitamin D3 (pre-D). The continued UV-irradiation causes isomerization of pre-D3 to D3, which is taken up by DBP. The presence of methyl group at C24 side chain in vitamin-D2 reduces its affinity for DBP (this is discussed in Section 2.3), resulting in faster clearance from the circulation. This factor limits the conversion of 1α,25(OH)_2_D2 to 25OHD2 by the CYP hydroxylases present in the system [45].

The subsequent steps in the biosynthesis, function, and regulation of 1α,25(OH)_2_D implicate a major group of enzymes, widely known as cytochrome P450 (CYP) enzymes or vitamin D hydroxylases. The key members of the CYP enzyme family, include CYP2R1, CYP27A1, CYP24A1, CYP2J3, and CYP3A4. In eukaryotes, they are membrane-bound and take part in biosynthetic reactions of steroid hormones and bile acid [46]. The initial step (Figure 2) occurs in the liver and also in some extrahepatic tissues such as the kidney. Vitamin D2 and D3 ingested orally or from the skin are metabolized to form 25-hydroxy vitamin-D2 and D3 (25(OH)D2 and 25(OH)D3), which is the circulatory form [47]. The multistep reaction is catalyzed by different members of CYP enzymes such as CYP2J3 and CYP3A4, in addition to CYP27A1 (a mitochondrial 25-hydroxylase) and CYP2R1. Both of the former enzymes are expressed in the liver and conserved among species and cause the generation of 25(OH)D by the addition of a hydroxyl group.

Consequently, a dominant target gene of 1α,25(OH)_2_D codes for CYP27B1 or 25-hydroxyvitamin D3 lα-hydroxylase, an enzyme found principally in the kidney, which metabolizes 25(OH)D to 1α,25(OH)_2_D [48,49]. 1α,25(OH)_2_D is the hormonally active form of vitamin D and is responsible for most of its biological functions. Various factors regulate the formation and inhibition of 1α,25(OH)_2_D. The appropriate signals such as the action of parathyroid hormone (PTH) promote the synthesis of 1α,25(OH)_2_D, while calcium, fibroblast growth factor-23 (FGF23), and phosphate, inhibit 1α,25(OH)_2_D [50]. CYP24A1 is induced by 1α,25(OH)_2_D and functions through a feedback mechanism to prevent vitamin D toxicity (VDT).

### 2.2. Vitamin D Metabolites

Clinicians find it difficult to quantify major metabolites of 1α,25(OH)_2_D found in circulation. Most of the vitamin-D intermediate metabolites can be epimerized with the help of epimerization enzymes like 3-epimerase [51]. These epimerization enzymes belong to the CYP group of enzymes and have the ability to further metabolize these epimers as in the common pathway [36]. Epimerization is a mechanism involved in vitamin D metabolism and occurs in extrarenal tissues [52]. Some of the common metabolites of vitamin D that are found in circulation along with their major functions have been described.

#### 2.2.1. 25-hydroxyvitamin-D2 (25(OH)D2)

The levels of 25(OH)_2_D2 (also known as 25-Hydroxyergocalciferol) produced by hydroxylation in the liver reflect an individual’s overall vitamin D status. Plants and mushrooms are important sources. Its deficiency has been linked to hypertension, autoimmune diseases, and cancer. It is a bone density conservation agent. Thus, it can inhibit bone resorption and aid in bone mineralization and regeneration. It is used to heal bone fractures and to treat bone diseases, such as osteopenia and osteoporosis [39].

#### 2.2.2. 25-hydroxyvitamin-D3 (25(OH)D3)

Obtained from the animal diet, 25OHD3 is transported to the kidney where it is hydroxylated in the mitochondria by 1-hydroxylase to form 1α,25(OH)_2_D3 and 24,25(OH)_2_D3. The deficiency of this metabolite is linked to hypertension, autoimmune diseases, and cancer [45].

#### 2.2.3. 24, 25-dihydroxyvitamin-D (24,25(OH)_2_D)

Vitamin D obtained from the diet goes through a series of biosynthetic reactions in the liver and kidney where it gets converted to either 24, 25-dihydroxyvitamin-D (24,25(OH)_2_D) or 1,25-dihydroxyvitamin D (1α,25(OH)_2_D) [53]. It has been reported that 24,25(OH)_2_D may have a role in bone mineralization [54]. Competition studies conducted by Sömjen et al. in (1982) demonstrated the existence of specific cytoplasmic and nuclear receptors (NRs) for 24,25(OH)_2_D D3 [55].

#### 2.2.4. 1-α, 25-dihydroxyvitamin-D3 (1α,25(OH)_2_D3 or 1,25D3)

1,25D3 is biologically active; has roles in calcium absorption and deposition; affects cellular differentiation and proliferation and central nervous system (CNS); and modulates immune responsiveness. Recent reports advocate it as a major chemopreventive agent against various malignancies including cancers of the stomach, prostate, and colon [56].

#### 2.2.5. 1-α, 25-dihydroxy vitamin-D2 (1α,25(OH)_2_D2 or 1,25D2)

Activated vitamin-D2 has been used in a broad range of studies to assess the immune function and calcium homeostasis. The ingestion of vitamin D_2_ results in an increase in serum concentrations of 1α,25(OH)_2_D2. Knutson et al., reported that an increase in 1α,25(OH)_2_D2 is accompanied by a comparable decrease in serum concentrations of 1α,25(OH)_2_D3 in rats and monkeys [56]

#### 2.2.6. 3-epi-vitamin D

3-epi-vitamin D denotes the following metabolites: 3-epi-25-hydroxyvitamin-D3 (3-epi-25(OH)D3); 3-epi-24; 25-dihydroxy-vitamin-D3 (3-epi-24,25(OH)_2_D3); 3-epi-1α,25-hydroxyvitamin-D3 (3-epi-1α,25(OH)_2_D3) [51]. Reddy and colleagues (2001) described an alternate metabolic pathway where C3 epimerization causes a change in the orientation of the hydroxyl group at position C3 of the A ring. 3-epi-vitamin D metabolites have less affinity to DBP and the VDR than the respective 25(OH)D3 and 1α,25(OH)_2_D3 forms [57]. 3-epi-1α,25(OH)D3 possesses some anti-proliferative and differentiation activities and suppresses PTH secretion but only at lower levels compared to the non-epimeric compounds [58]. Recent publications have shown the presence of 3-epi-vitamin D in infant, pediatric, and adult populations [37].

Further studies are required to ascertain the source of epimers, whether they are attained exogenously or formed endogenously and to determine their specific roles. Since 1α,25(OH)_2_D3 extracts are sensitive to photodegradation, the quantification of epimers present in vitamin D supplements is difficult.

### 2.3. Vitamin D Transport

A minor portion of the free or unbound 1α,25(OH)_2_D enters the cells through simple diffusion. The majority of circulating vitamin D and its metabolites are attached to special carrier proteins called DBP or albumin. 1α,25(OH)_2_D-DBP complex enters the cell with the help of membrane proteins like caveolin, cubilin, and megalin or through simple diffusion (Figure 3) [59]. DBP, which is synthesized in the liver is a steroid binding protein and acts as a carrier molecule of 25(OH)D [20,60,61]. It has a higher affinity to 25(OH)D and 24,25(OH)_2_D than vitamin D or 1α,25(OH)_2_D. DBP acts as a storage protein that accumulates 25-OHD under conditions of accessibility to 1α,25(OH)_2_D and usually releases it for biological requirements [62]. In patients suffering from liver diseases, like specific protein-losing enteropathies or the nephrotic syndrome, the synthesis of normal levels of DBP and albumin may be affected. These individuals may have low total levels of DBP, resulting in the presence of unbound vitamin D metabolites in the serum [63].

Vitamin-D2 and D3-DBP complexes are internalized by the liver. The hepatocytes convert the vitamin D into 25(OH)D. DBP-25(OH)D is internalized by a process called receptor-mediated endocytosis in the renal tubules of the kidney. Megalin and cubilin proteins help in receptor-mediated endocytosis and occur in the proximal convoluted tubules of the kidney and in extra-renal tissues, like mammary cells [64]. Figure 3 shows the critical steps in vitamin D activation taking place in the renal cell. 25(OH)D dissociates from DBP and gets hydroxylated to 1α,25-(OH)_2_D by CYP27B1. Subsequently, CYP24A1 (24-hydroxylase) can act on two substrates, 1α,25(OH)_2_D and 25(OH)D through negative feedback mechanisms and convert them to inactive end-products like 1,24,25-trihydroxyvitamin D (1,24,25-(OH)_3_D) and 24,25-dihydroxyvitamin D (24,25-(OH)_2_D), the roles of which are discussed in Section 5 [65].

## 3. Vitamin D Receptors (VDRs)

Maestro et al., (2016) reported that the vitamin D receptor (VDR) is found in most human tissues and has more than 1000 target genes [41]. 1α,25(OH)_2_D3 exerts its effect with the help of high-affinity VDR through a series of cell-signaling reactions or as a ligand-activated transcription factor [66,67]. The mode of action may be genomic or non-genomic and can be executed using nVDR or mVDR, respectively [68,69]. In a cell, 1α,25(OH)_2_D exerts its non-genomic effects by initiating multiple signaling pathways via binding to a membrane receptor. These result in immediate responses in the target cells. However, 1α,25(OH)_2_D3 exerts its genomic effects through the NR, which binds to target genes and leads to gene expression [70]. Detailed discussions on the distinguishing features of the two types of receptors, the nature of their diversity, and functions are provided in the upcoming sections.

### 3.1. Nuclear VDR (nVDR)

The VDR is widely distributed and not limited to the classic target tissues of vitamin D. The VDR belongs to the superfamily of NRs. It acts as a transactivating transcription factor by regulating the expression of genes that mediate its biological activity [71]. The 1,25D-dependent genomic responses in the nucleus materialize as a result of the ligand-activated transcription factors and are relatively slow (may take from several hours to days) [72].

In humans, the VDR is encoded by a large gene (>100 kb), which is present on the long arm of chromosome 12q12. It possesses a complex intron/exon structure and is composed of nine exons (Figure 4) [66]. Exons 1a to 1f encode the 5′ untranslated region, UTR of the VDR mRNA and are alternatively spliced. Exons 2 and 3 encode the translation start site, and consists of the DNA-binding domain. This region comprises two highly conserved zinc-finger motifs, one in each domain and is a distinguishing feature of the NRs. Exons 4 to 9 encode the overlapping ligand-binding domain, LBD as well as strong heterodimerization domains. Exon 9 localizes at the 3′ UTR region [71]. The *Vdr* gene of a mouse is located on chromosome 15 [72].

The DNA binding domain of the VDR contains a zinc-finger motif which is the core of the VDR protein, involved in the activation of the target genes. Upon binding of 1α,25(OH)_2_D3 to the receptor, two independent protein interaction surfaces form on the VDR protein, one of which is needed for specific DNA binding, while the other recruits the large coregulatory complexes and is necessary for gene modulation [58].

Subsequently, the VDR forms a heterodimer with the retinoid X receptors, RXR (Figure 5). This heterodimer recognizes the vitamin D response element (VDRE) located in the promoter of vitamin D target genes. The 12-helical LBD is located in the C-terminal region of the receptor and binds to 1α,25(OH)_2_D3 [73]. The coregulator proteins complex with 1α,25(OH)_2_D3 to activate (coactivators) or inhibit (corepressors) the target gene transcriptional activity. The coactivators mainly include the following two types: (i) Proteins like the vitamin D receptor interacting protein complex, DRIP, also known as a mediator, form a bridge that connects VDR binding to VDRE at the transcription start site (TSS) and (ii) Proteins such as the steroid receptor coactivator family 1-3 (SRC 1-3) help in transactivation [17]. *VDR* trans-repression involves diverse sets of mechanisms involving the suppression of gene expression by various transcription factors [74,75]. Genes get transcribed, leading to the formation of respective proteins like SLUG, which may downregulate genetic functions by the VDR [76].

The target genes of vitamin D respond in a cell-specific manner. It has been discovered that the total number of vitamin D targets in the human genome is more than one thousand. This is further supported in cells such as human lymphoblastoid and THP-1 human monocytic leukemia cells used for chromatin immunoprecipitation combined with high throughput sequencing (ChIP-seq) [77,78]. Table 2 shows some of the target genes of 1α,25(OH)_2_D, along with their specific roles and functions [42,79,80,81].

X-ray crystallographic studies have shown that the nVDR-1α,25-(OH)_2_D complex has a classic genomic ligand binding pocket called the VDR-genomic pocket, GP and an alternative ligand binding pocket, VDR-alternative pocket, AP, that partially overlaps the VDR-GP [81]. Rochel et al. (2000) demonstrated that the VDR-GP favors a bowl-like molecular geometry with 1α,25(OH)_2_D3 by X-ray and computational studies [82]. This shape initiates genomic responses, whereas the VDR-AP accepts a ligand with a more planar-like, extended molecular geometry and initiates a rapid response [82].

The *nVDR* gene is frequently susceptible to polymorphisms. Polymorphism can be defined as subtle sequence variations transpiring in a population and are detected using techniques like restriction fragment length polymorphisms (RFLPs). It is a technique that exploits variations in homologous DNA sequences. Polymorphisms alter gene expression, thereby affecting protein levels of the *VDR* gene, leading to functional changes [83,84]. The research data indicate that *VDR* gene polymorphisms are associated with diabetes; cancers of the colon and prostate; autoimmune disorders like Crohn’s disease, Grave’s disease, and multiple sclerosis; kidney diseases; and various neurodegenerative diseases like schizophrenia [85]. The actions of 1α,25-(OH)_2_D3 are most likely due to the combined effects of both genomic and non-genomic responses [86].

### 3.2. Membrane VDR (mVDR)

The rapid, non-genomic activity is affected by the interaction of 1α,25(OH)_2_D3 ligand with specific binding sites on the plasma membrane and vesicle membranes of target cells referred to as membrane VDR (mVDR). mVDRs are also known as 1,25 membrane-associated rapid response steroid-binding proteins (1,25-MARRS), protein disulfide isomerase family A member 3 (Pdia3), endoplasmic reticulum proteins 57 and 60 (ERp57, ERp60), and glucose-regulated protein 58 (GRP58) [87,88]. These complexes are thought to be modified VDRs. They are chaperone proteins and coupled to signal transduction systems [87]. In this review, the mVDRs are addressed by the name 1,25-MARRS.

Chick intestinal cells; adult rat and mouse cardiomyocytes, epithelial cells, myocytes, osteoblasts, chondrocytes, and cancer; and in humans, bone, teeth, brain, parathyroid, muscle, kidney, and cancer cells are 1,25D-responsive cells that function via the rapid response mechanism [88,89]. Some of the widely studied responses include calcium uptake from the intestine, insulin secretion by β-cells of the islets of pancreas, the growth and differentiation of smooth muscle cells, deviations in chondrocyte and keratinocyte growth, the changes in intracellular second messengers including Mitogen-Activated Protein Kinase (MAPK), phosphoinositides and cyclic AMP (cAMP), as well as the opening of the voltage-gated ion channels and the activation of cytosolic kinases [47,90]. Nevertheless, the rapid genomic actions are less studied, and the downstream responses following ligand-receptor binding are obscure.

1,25-MARRS are associated with membrane scaffold proteins like caveolin that are associated with lipid-rich craters of the plasma membrane and are responsible for the formation of caveolae in multiple cell types [91]. They are associated directly with the signaling pathways of the cells, using a rapid response that is elicited in the cell by the activation of the receptor complex upon ligand binding (Figure 4) [92]. Garbi et al. (2006) reported that ubiquitous ERp57 deletion in mice is embryonically lethal [93]. Wang et al. (2010) developed *Pdia3*-deficient mice to study the role of skeletal development [94]. It was revealed that no homozygous mice were observed at birth indicating that the *Pdia3* gene knockdown resulted in lethality of the homozygous mice at the embryonic stage. There were abnormalities in the skeletal tissues of *Pdia3^+/−^* mice compared to the *Pdia3^+/+^* mice [94]. In chick intestinal epithelial cells, targeted disruption of the 1,25-MARRS receptor gene was observed to eliminate the rapid response to 1α,25(OH)_2_D3 in terms of rapid calcium uptake and Protein kinase A (PKA) signaling [95]. An association between the nVDR and 1,25-MARRS was observed in 1α,25(OH)_2_D3-mediated growth regulation of rat growth plate chondrocytes. It was also proven that chondrocytes from *nVdr*^(−/−)^ mice have Pdia3 and exhibit a rapid escalation in Protein kinase C (PKC) levels in response to 1α,25(OH)_2_D3 [96]. These findings suggest that 1,25-MARRS has other functions in addition to acting as a receptor for 1α,25(OH)_2_D3.

Caveolae are flask-shaped invaginations found on the plasma membrane and other membranes of cellular organelles, the main components of which are cholesterol, sphingolipids, and the protein, caveolin [97]. The structural components of caveolae are caveolin and exist as three isoforms, caveolin-1, -2, and -3, according to their location and function in different cells. Caveolin-1 and -2 are the most extensively studied [98]. The primary function of caveolae was thought to be vesicle trafficking, but subsequent studies have shown that they are significant partakers in cholesterol biosynthesis, signal transduction, tumor suppression, and anti-proliferative action. Caveolin-1 is found to interact with and regulate the receptor protein signaling and cellular docking mechanisms. Upon activation, they get endocytosed and translocated to the nucleus where they exert their genomic function. Recently, a loss of caveolin-1 was identified as a marker of the cancer-associated fibroblast phenotype. Caveolae are found to contain mVDR, which indicates the vital role of caveolae and caveolin proteins in VDR signaling on cell membranes [99].

Studies have linked components of cell-signaling pathways like diacylglycerol (DAG), phospholipase A2 (PLA2), and phospholipase A2-activating protein (PLAA) to mVDR, which may point to the role of caveolae in the signaling axis of 1α,25(OH)_2_D3 and 1,25-MARRS by interacting with G proteins [87].

## 4. Signaling Pathways of Vitamin D

As mentioned earlier, recent studies have established that vitamin D and associated metabolites can evoke a rapid, non-genomic response in the cell by binding to 1,25-MARRS [87,90,91,99,100]. Numerous studies on 1,25-MARRS activated signaling pathways over the past three decades indicate that the major pathway was related to calcium influx, and is termed transcaltachia [101]. This review is focusing on a few of the most common and widely discussed signaling pathways like regulation of PKC-mediated calcium and phosphorus uptake [102], phosphatidyl inositol (PI) and PLA2 activity [103], and MAPK pathway [104] and how they are involved in genomic cross-talk (Figure 5). However, there are several other signaling pathways like PKA/cAMP, protein kinase B [104], CAMK, and multiple MAPK pathways [105] that are activated by 1,25-MARRS. Detailed reports on how the various signaling pathways are affected by 1,25-MARRS can be found elsewhere [106].

PKC activation is observed in chondrocytes by binding of 1α,25(OH)_2_D3 to 1,25-MARRS [111]. The receptor-ligand binding results in a rapid increase in the amount of PLA2 activity. PLA2 activates PKC with the help of several intermediate products, the key one being arachidonic acid. Arachidonic acid can directly activate PKC, while serving as a substrate for several other enzymes. Lysophospholipids are intermediates of arachidonic acid release. They are capable of activating phospholipase C, which hydrolyzes PI to DAG and IP3. IP3 binds to receptors that are located on the smooth endoplasmic reticulum, causing Ca^2+^ release, which acts as a co-factor for PKCα. DAG binds to PKCα, activating the enzyme and translocating it to the plasma membrane [112]. Richard et al. first demonstrated the involvement of 1,25-MARRS in anti-cancer activity in breast cancer cells [101]. 1α,25(OH)_2_D3 is thought to inhibit the phosphorylation and expression of Akt, a kinase responsible for regulating cell survival and proliferation [113]. 1α,25(OH)_2_D3 promotes angiogenic factor interleukin-8 (IL-8) in squamous carcinoma cells, but calcitriol interrupts IL-8 signaling in prostate cancer cells and inhibits the migration and tube formation of endothelial cells [114,115]. Hilliard and coworkers in 1994 reported that treatment of COS-1 cells with 1α,25(OH)_2_D3 caused an increase in the phosphorylation of serine residue 208 (S208) located in the VDR, thus recruiting the coactivator to form the nVDR-RXR heterodimer [116]. On the other hand, antagonistic functions of genomic and rapid response pathways have been observed in breast cancer cells [117].

Cross-talk exists between the genetic and rapid signaling pathways. This is brought about by phosphorylation of the chief proteins in the VDR-transcriptional complex and is often observed in cell-cycle regulation. The different modes of signaling actions and cross-talk operated by 1α,25(OH)_2_D3-1,25-MARRS association lead to transcriptional modulation of several genes, depending on the cell- or tissue-type considered. Barletta and coworkers (2002) reported that 1α,25(OH)_2_D3-1,25-MARRS mediated rapid activation of cytosolic kinases and may be responsible in phosphorylating key coactivators involved in diverse pathobiological and physiological processes [118]. The diverse biological effects result from genomic or non-genomic cross-talk to some extent as well as through context-specific functions of the target genes. Certain second messengers may engage in cross-talk with the nucleus to regulate gene expression, particularly the Rapidly Accelerated Fibrosarcoma/Mitogen-Activated Protein Kinase (RAF/MAPK) pathway [119]. Deeb et al. (2007) reported the activation of MAPK–ERK (extracellular signal-regulated kinase) 1 and 2 cascades through the phosphorylation and activation of RAF by PKC, partly induced by the changes in intracellular Ca^2+^ concentration [120].

## 5. Regulation of Calcium and Phosphate Homeostasis by 1α,25(OH)_2_D3 and PTH

An essential function of 1α,25(OH)_2_D3 is the maintenance of serum calcium and phosphate levels. Normal serum Ca^2+^ and PO_4_ levels serve as the cornerstones for several biological processes. 1α,25(OH)_2_D3 helps in effective utilization of calcium by enhancing intestinal Ca^2+^ absorption or Ca^2+^ mobilization from bones in the absence of calcium in the diet. It acts together with the parathyroid hormone (PTH) to help maintain mineral homeostasis between the blood and bones, the neuromuscular and immune function, and cardiovascular health, and is essential in the suppression of malignant changes [121].

Plasma Ca^2+^ and PO_4_ concentrations are maintained at constant levels of 2.2–2.6 mMol/L and 0.8–1.45 mMol/L, respectively [121]. A drop in serum Ca^2+^ levels stirs a series of quick reactions in the biological system, illustrated in Figure 6.

The parathyroid glands possess transmembrane calcium-sensing proteins that are coupled to a G protein system and detect the levels of Ca^2+^ in the plasma [122]. This stimulates the secretion of PTH [123]. PTH advances to the osteoblasts in bones and the proximal convoluted tubule cells in the kidney within seconds. PTH stimulates the mobilization of Ca^2+^ from the bones and reduces its excretion by the kidney. These brisk effects increase the concentration of Ca^2+^ in the blood. In conjunction with these changes, PTH promotes the activity of the CYP27B1 gene in the proximal convoluted tubule cells of the kidney [124]. Therefore, the proximal convoluted tubule cells of the kidney act as the endocrine gland of the vitamin D hormone [125]. 1α,25(OH)_2_D3 has a quintessential role in Ca^2+^ level homeostasis. 1α,25(OH)_2_D3 signals the intestine to augment Ca^2+^ absorption and in association with PTH, promotes mobilization of bone Ca^2+^ and PO_4_ and renal reabsorption of minerals. Consequently, this leads to the deposition of more Ca^2+^ and PO_4_ in the system, which boosts both mineral levels in the serum [122]. When the levels of serum Ca^2+^ exceed the set point of the calcium-sensing system, the parathyroid gland-induced cascade of events is shut down. The C-cells present in the thyroid gland secrete calcitonin in response to the high serum Ca^2+^ levels. Calcitonin is a 32-amino acid peptide that blocks Ca^2+^ mobilization from the bone [126]. Calcitonin also stimulates the renal CYP27B1 to provide 1α,25(OH)_2_D3 for non-calcemic needs under normal Ca^2+^ conditions [127].

When the Ca^2+^ levels return to normal, the 24-hydroxylase enzyme in the kidney is stimulated, which breaks down the active 1α,25(OH)_2_D3 to its inactive form. The enzyme also breaks down 25(OH)D3, which is the form of vitamin D that is stored in the body.

The increase in the synthesis of 1α,25(OH)_2_D3 in response to hypocalcemia (via the secretion of parathyroid hormone) occurs within a few hours, whereas the parathyroid glands respond to hypocalcemia within a few minutes. Under hypercalcemic conditions, the secretion of PTH is suppressed, and the synthesis of 1α,25(OH)_2_D3 is significantly reduced, whereas the synthesis of another inactive metabolite, 24R, 25-dihydroxyvitamin D3 (24(R),25-(OH)_2_D3), destined for excretion is increased [128].

## 6. Vitamin D and the Gastric Tissue

Notably, in the last four decades, non-calcemic effects of 1α,25(OH)_2_D3 have been explored and investigated. Studies have revealed that 1α,25(OH)_2_D3 as well as its metabolites play a key role in controlling and regulating genes responsible for preserving the integrity of the epithelial barrier including the GI tract, in addition to immunoregulatory and inflammatory responses [18]. One of the best studied non-traditional functions of 1α,25(OH)_2_D3 is the cellular mechanisms involved in tumor control and repression of proliferating epithelial cells of the skin [129], breast [130,131], colon [132], and prostate [133]. Therefore, the majority of the studies have focused on tumor cells and their progenitors along with the tumor microenvironment (TME) [134].

There is a need for further research to study the roles of 1α,25(OH)_2_D3 and its metabolites in maintaining the equilibrium and stability of normal tissue. Despite the increase in the number of studies, publications, and summits centering on the various utilities of 1α,25(OH)_2_D3, there is little awareness regarding the metabolites and precise functions of vitamin D in the gastric environment [135]. Scientists, clinicians, and dieticians have started to explore the therapeutic advantages of this steroid hormone in gastric cancer (GC) coupled with its role in the TME.

This review is an attempt to assemble a few of the important articles that presents information regarding the regular expression and function of vitamin D and the VDR in the gastric tissues.

Furthermore, the authors were interested in studying the role of vitamin D and the VDR in the normal proliferation and differentiation of the gastric cells as well as in maintaining the cellular and physiological homeostasis of the gastric tissues. A systematic literature review was conducted using the electronic search engines and online databases like Cochrane Library, Google Scholar, PubMed Central, Semantic Scholar, Scopus, ScienceDirect, SciFinder, and RefSeek. The databases were explored for articles published in the English language between the years 1927 and 2019, using the keywords, vitamin D, calcitriol, stomach and gastric environment and gastric tissue. However, the articles focusing solely on the role of vitamin D and its analogues in gastric progenitor cell proliferation and the differentiation and formation of gastric epithelium were challenging to acquire. Nevertheless, a few articles that are related to this area to some extent are included in the study. This endeavor warrants the scrutinization of the central role of the VDR in 1α,25(OH)_2_D3 action and the therapeutic roles of 1α,25(OH)_2_D3 and its analogs in maintaining gastric tissue homeostasis and controlling tumor proliferation, and thereby in the betterment of human health. Table 3a,b indicate the major available studies that emphasize critical findings in the human and animal stomachs. Table 3a,b indicate the major available studies that emphasize key findings in the human and animal stomachs.

One of the first studies regarding 1α,25(OH)_2_D in the stomach was conducted by Leigh Claire in 1927. The study attempted to explicate the source of 1α,25(OH)_2_D in the oil present in the stomach of fulmar petrel (*Aestralata lessoni*), also known as muttonbird. The main diet of fulmar petrel and Australasian cod includes whale feed which consists principally of a bright red crustacean. It was earlier thought that the stomach oil was a residue from the digestion of whale feed, but an analysis of the stomach oil showed an abundance in cetyl esters, which are relatively absent in whale feed. The study showed that rats fed with muttonbird stomach oil were cured of rickets. This confirmed the presence of 1α,25(OH)_2_D in the oil. However, the source of 1α,25(OH)_2_D in the stomach oil remained unknown, even though it was suggested that whale feed is the common source of 1α,25(OH)_2_D in stomach oil and cod-liver oil [136].

The field received a spur when the toxic effects of 1α,25(OH)_2_D3 were described in various publications as early as 1928 [138]. Hypervitaminosis of 1α,25(OH)_2_D3 causes pathological modifications such as calcification in the skeletal and soft tissues like the stomach and kidneys [167]. One such study published by Dewind in 1961 stated that excessive ingestion of 1α,25(OH)_2_D3 might contribute to osteosclerosis [168]. The pathological effects of ergocalciferol on gastric tissues were reported by Selye and Bois (1957). They demonstrated that a subcutaneous injection of ergocalciferol simultaneously with cortisol acetate in female Sprague-Dawley rats caused calcification of the tissues of the gastric mucosa, but steroids like deoxycorticosterone acetate inhibited this effect [137]. Another study in albino rats treated with 1α,25(OH)_2_D3, dihydrotachysterol (ATl0) and parathyroid extract showed that overdosage of these substances could lead to metastatic calcification of the stomach tissues as well as the gastric mucosa [148].

Kimura et al. (1967) reported an autopsy case of hypervitaminosis D where the patient died due to calcium deposition in bones and soft tissues including the stomach and esophagus, subsequently leading to uremia [151]. Subsequently, several researchers have studied the toxic effects of hypervitaminosis D and hypercalcemia in soft-tissue mineralization (STM), which induces lesions in the proximal and cortico-medullary regions of the renal tubules, aortic media and intima, myocardial arterioles and myofibers, and gastric glandular mucosa and muscularis [139]. The 1960s witnessed fundamental research on the synthesis, metabolism, and clinical relevance of 1α,25(OH)_2_D3. It was found that a deficiency of 1α,25(OH)_2_D3 can lead to hepatic and gastrointestinal diseases that ultimately affect the bone [84].

The VDR was discovered in 1969, and it was established that the VDR was a cell nucleus-localized receptor [169]. It was found that the VDR is present in over 30 target tissues in humans comprising the bone, cartilage, intestine and kidney, as well as in pancreatic β-cells [170], hair follicles [43], various cancer cells [119], and B and T lymphocytes [171]. With the advent of new techniques, Stumpf and coworkers (1979 and 2008) and Holick (1995) established that rat duodenum, jejunum, ileum, colon, stomach [143], pituitary, parathyroid, kidney [172], and skin are target tissues of 1,25(OH)_2_D3 [147]. Autoradiographic studies by Stumpf et al. (1995) indicated the nuclear binding of radioactively labeled 1,25(OH)_2_D3, and its analog, 22-Oxacalcitriol, in neck mucus cells of gastric and pyloric glands and dispersed endocrine cells in the antrum [144]. Moreover, the *Vdr* gene expression was found in the oxyntic mucosa of rats [146]. These discoveries assisted in correlating a 1α,25(OH)_2_D3 deficiency and fat malabsorption to partial gastrectomy [82,173,174]. Gastric bypass surgery (GBP) or sleeve gastrectomy is a common and widely advised medical treatment (surgery) to overcome morbid obesity by achieving significant weight loss and improving associated medical conditions [175]. Despite the beneficial effects of GBP, the complications such as nutritional deficiencies, metabolic bone disease (MBD), and osteoporosis have been reported. Post-gastrectomy MBD is a distinct condition associated with gastrectomy, due to calcium and 1α,25(OH)_2_D3 malabsorption, and includes subclinical osteomalacia symptoms [176,177]. A possible explanation provided by Axelson for this trend is that an escalation of serum 1α,25(OH)_2_D levels in such patients may enhance the ability of the small intestine to absorb calcium [142]. Rino et al. points to less food intake and fat malabsorption to be potential causes of the deficiency of fat-soluble vitamins in general post-gastrectomy, but no significant reduction in serum vitamin D levels was seen [178]. Several case studies have reported attempts to examine and investigate the pathophysiology of osteomalacia and osteoporosis following gastrectomy [150,152].

The scientific world witnessed novel endeavors to uncover the numerous mechanisms of 1α,25(OH)_2_D3 responsible for maintaining gastric homeostasis. A major objective is to unearth a relationship between calcium or phosphate homeostasis and gastric secretions, like gastric acid and gastrin hormone. Gastric secretions and homeostasis are maintained by highly meticulous and precise mechanisms [179]. Acid-secretion in the stomach is guarded by a well-schemed machinery comprising hormonal, neuronal, and paracrine modules. A remarkable study in this subject was conducted by Selking O. et al. (1982), which indicates that hypercalcemia does not cause an increase in gastrin secretion in rats, and further reinforces previous findings [140,150,152,179]. However, follow-up investigations revealed more than a few unpredictable findings. Kurose, T. et al. (1988) and Gagnemo-Persson (1999) demonstrated that calcium and 1α,25(OH)_2_D3-deficiency impairs gastrin and somatostatin secretion in the perfused rat stomach [141,146,180].

Enterochromaffin-like (ECL) cells constitute one of the largest endocrine cell populations in the mammalian system. They are controlled by the gastrin hormone and are rich in histamine, calcium-binding protein calbindin, and chromogranin A. Gastrin and the ECL cells form a functional unit, known as the gastrin–ECL-cell axis [181]. Gagnemo-Persson (1999) reported that 1α,25(OH)_2_D3 or 1α,25(OH)_2_D3-dependent mechanisms play a direct role in suppressing the gastrin–ECL-cell axis [146]. Kopic and Geibel (2013) investigated the effect of calcium on gastric physiology. Serum calcium levels are controlled by the combined actions of PTH and 1α,25(OH)_2_D3 [158]. The minute variations in calcium homeostasis can have adverse effects. The homeostasis is maintained by a highly sensitive calcium-sensing receptor (CaSR) present on the parathyroid gland in addition to tissues, like the kidney and gastrointestinal tract (GIT) along with a negative feedback loop by 1,25(OH)_2_D3 [158]. In humans, calcium stimulates gastric acid secretion. In diseased states like hyperparathyroidism, PTH influences gastric cells and gastrin release with the aid of calcium [158,182]. The studies revealed controversial results regarding the roles of 1,25(OH)_2_D3 and its impact on gastric physiology.

Studies have demonstrated an association between low levels of 1α,25(OH)_2_D and related complications in people with highly melanized skin. Harris in 2006 reported that a high percentage of American black people have lower serum 25(OH)D3 than their white counterparts [183]. Giovannucci et al. (2006) conducted a study in a cohort of black and white male health professionals that were relatively homogenous in education and living standards [153]. The results indicated that black people are at a higher risk of digestive system cancers associated with hypovitaminosis D, and thus a higher mortality rate [153]. Several studies have hypothesized that the nutritional status and serum levels of 1α,25(OH)_2_D3 might influence the chances of GC. Chen et al., (2007) observed that higher concentrations of serum 25(OH)D3 intensifies the hazard of esophageal squamous cell carcinomas in males but not in females, but no association was found in the case of the gastric cardia or non-cardia adenocarcinoma in either sex [154].

Gastric ulcers are caused by a disruption in the homeostasis between the acidic and protective environment of the gastric mucosa [161,184]. The ulceroprotective and antioxidant effects of 1α,25(OH)_2_D3 were reported in rats [149]. *H. pylori* gastritis infection, which is the Th1 type, causes autoimmune gastritis (AIG), and is classified as an autoimmune disease. The low serum 1α,25(OH)_2_D3 status in *H. pylori* gastritis patients could possibly act as a predisposing factor for the severe Th1-type AIG, which causes intense damage to the stomach epithelium [156]. The gut microbiome is reactive to 1α,25(OH)_2_D3 supplementation and deficiency despite considerable heterogeneity [164,165]. For example, 1α,25(OH)_2_D3 could boost the intracellular killing of the replicating *H. pylori* bacteria [159,185]. Fletcher et al., (2019) reported the importance of 1α,25(OH)_2_D3 in maintaining the gastrointestinal barrier integrity, surveillance of the gut microbiota, and inflammatory immune responses [166].

Du et al. reported that 1α,25(OH)_2_D3 and the VDR interaction can induce a cascade of gene regulation and cell signaling reactions to promote anti-tumor mechanisms [163]. Vitamin D and its analogues play an important role in reducing the risk of gastric adenocarcinomas. Expanding evidences have indicated that prevention of vitamin D deficiency is an economic and promising way to reduce risk of GC [157,162]. In 2015, Wen et al., studied the expression of local VDR in normal, premalignant, and malignant gastric tissues by immunohistochemical techniques [160]. The relationship between the VDR and clinicopathological factors of GC patients was also analyzed. They reported that the VDR was dynamically expressed in various types of gastric tissues. On treatment with 1α,25(OH)_2_D3, a decline in the VDR expression was observed from pre-malignant to gastric cancer tissues, and the VDR expression was the least in poorly differentiated tissues [160]. This shows that the VDR can be a prospective prognostic factor for GC. Ikezaki (1996) reported that the development of atypical hyperplasias and adenocarcinomas was significantly diminished in the glandular stomachs of rats by exposure to 24R,25(OH)_2_D3, which shows that 24R,25(OH)_2_D3 has chemopreventive effects [145].

Epigenetic mechanisms like chromatin remodeling complexes, DNA methylation, miRNA, and histone modifications are associated with GC [186]. Chromatin remodeling is capable of altering the chromatin structure, which disrupts the competence of regulatory proteins from gaining access to the target sites. Histone modifications, like acetylation, are vital in transcription and are synchronized by a balance between the different activities of histone acetyltransferases and histone deacetylases (HDACs) [187]. This process thereby amends the interaction of regulatory proteins with their targets and correlates with the transcriptional status of the genes [188]. Trichostatin A (TSA) is a known potent inhibitor of HDACs. TSA reportedly has therapeutic effects as it inhibits HDACs with zinc-containing catalytic sites, leading to the accumulation of acetylated histones in the nucleus and subsequent activation of target genes [189]. DNA methylation involves the addition of a methyl group to cytosine residues by enzymes called DNA methyltransferases (DNMTs). It is associated with transcriptional silencing. The presence of 5-Aza-2′-deoxycytidine (5azadC), a cytidine analog in DNA, prevents DNA methylation [155,190,191]. In association with TSA/sodium butyrate (TSA/NaBu) and 5-aza-2-deoxycytidine (5-Aza), 1α,25(OH)_2_D3 can induce apoptosis in GC cells through the expression of phosphatase and tensin homolog deleted on chromosome 10 (PTEN), which is an important tumor suppressor gene. This can lead to the potential application of 1α,25(OH)_2_D3 as a novel molecular target in GC therapies in association with the usage of TSA ⁄ NaBu and 5-Aza [155].

## 7. Conclusions

Unfortunately, for the gastroenterologist, there are limited data available on the impacts of VDD associated with gastric disorders. Considering the main roles of 1α,25(OH)_2_D3 in the immunoregulatory and preservation of the morphology and physiology of tissues, it is worth investigating the importance of 1α,25(OH)_2_D3 and its metabolites in maintaining stomach homeostasis. The information from research can be used for public awareness, recommending it as a necessary dietary supplement in daily lives thereby, improving human health. In the past 30–40 years, researchers have revealed an in-depth and detailed mechanism of how this miracle hormone facilitates various physiological processes in many biological systems. However, extensive research, both basic and applied, is required to explain the role of 1α,25(OH)_2_D3 and its metabolites in the gastric environment. If scientists, and the agencies that fund them, direct their efforts and expertise to focus on the role of this vitamin, better ways to treat abnormalities arising in the gastric tissues can be anticipated. These attempts can make more efficacious and serious use of 1α,25(OH)_2_D3, its metabolites, and synthetic analogs, which can cause a significant paradigm shift in the progress of medical science and the welfare of human and animal populations.

## Figures and Tables

**Figure 1 ijms-20-03832-f001:**
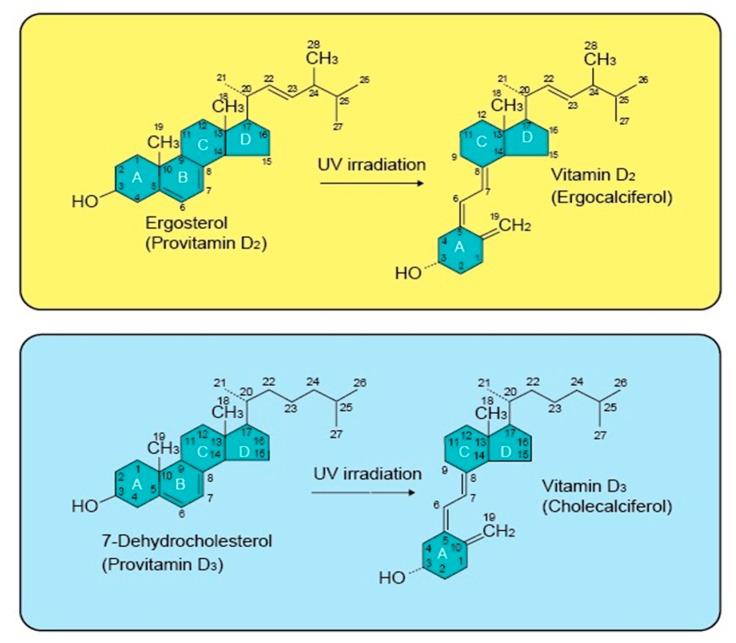
The structure of vitamin D2, vitamin D3, and their precursors. The structural difference between vitamin D2 and D3 is present in their side chains. The side chain of vitamin D3 has a broken ring, while D2 contains a double bond between carbons, 22 and 23, and a methyl group on carbon 24 on the broken ring.

**Figure 2 ijms-20-03832-f002:**
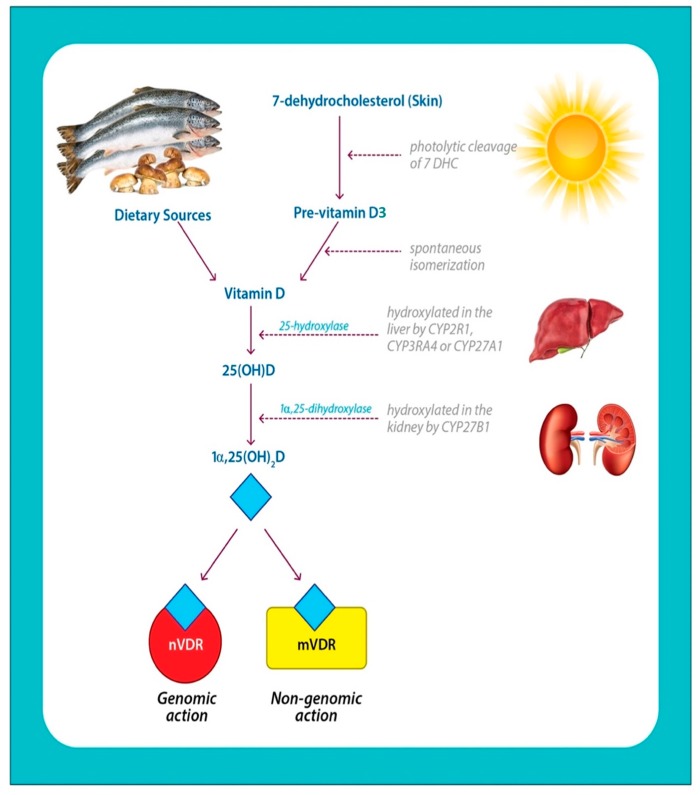
Photobiosynthesis and activation of Vitamin D. 7-dehydrocholesterol in the skin is converted to pre-vitamin D3, upon exposure to sunlight, which contains UVB radiation. Pre-vitamin D3 is converted to vitamin D3 in a heat-dependent process. Vitamin D2 and D3 from the diet are incorporated into chylomicrons and introduced into the circulation. Vitamin D formed in the skin or ingested through diet is stored and released from fat cells. In the serum, vitamin D is circulated while being bound to the vitamin D binding protein, DBP. DBP transports it to the liver where it is converted to 25(OH)D (circulating form) by vitamin D-25-hydroxylase. This form of vitamin D is biologically inactive and is converted in the kidneys to the biologically active form, 1α,25(OH)_2_D by 25-hydroxyvitamin D-1α- hydroxylase (1α-OHase). The 1α,25(OH)_2_D binds to the membrane vitamin D receptor (mVDR) or the nuclear vitamin D receptor (nVDR) and elicits specific biological responses.

**Figure 3 ijms-20-03832-f003:**
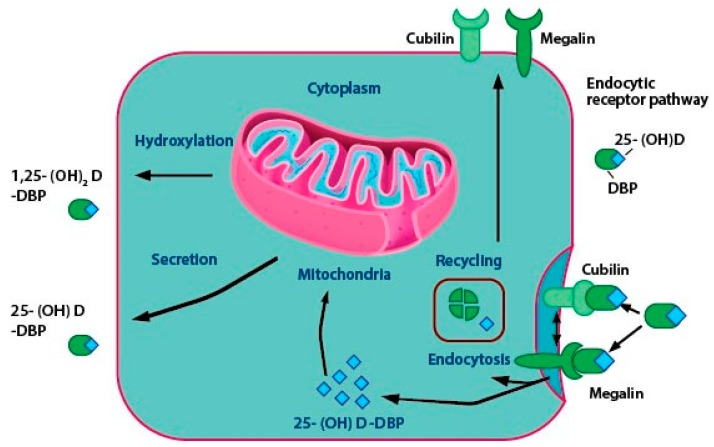
The roles of endocytic proteins in the delivery of 1α,25(OH)_2_D in the renal cells. The majority of the circulating 25-hydroxyvitamin D is bound to DBP, which is endocytosed via megalin and cubulin-mediated endocytosis. DBP is degraded, and 25(OH)D is either converted to 1α,25(OH)_2_D in the mitochondria for CYP27B1-mediated bioactivation or is secreted into circulation where it binds to DBP by CYP24A1-mediated inactivation. Cubilin and megalin then return back to the cell surface and the process gets repeated.

**Figure 4 ijms-20-03832-f004:**
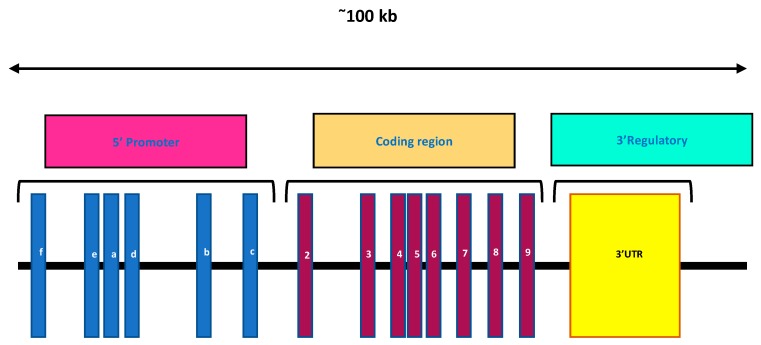
A map of the *nVDR* gene on chromosome 12q12. Blue boxes: Exon 1 (a to f), maroon boxes: Exons 2 to 9, yellow box: 3′ UTR.

**Figure 5 ijms-20-03832-f005:**
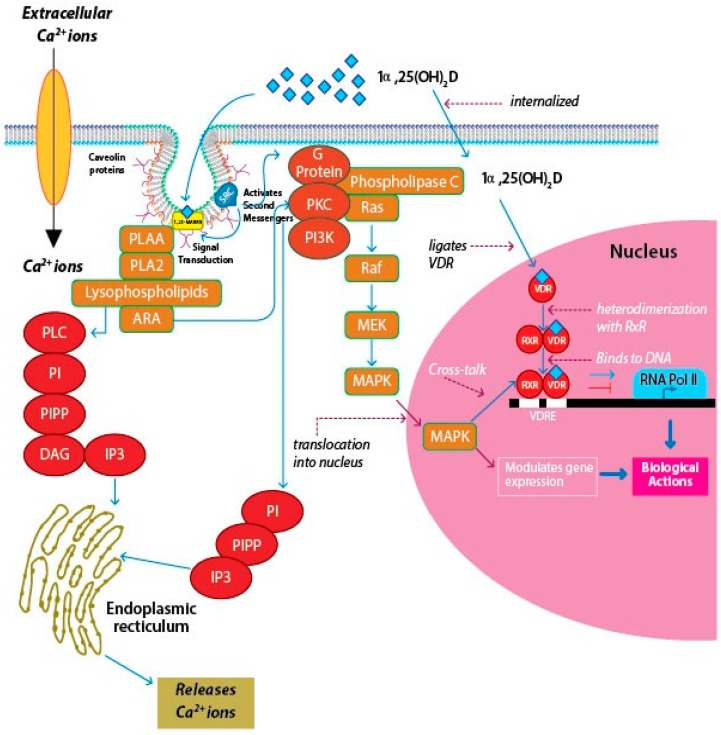
The mechanism of action of membrane VDR and nuclear VDR. On binding of the appropriate ligand to mVDR, cellular signal transduction systems that are linked to the membrane receptor get activated, which in turn, trigger the second messengers, resulting in a rapid response. The 1α,25(OH)_2_D3 binds to the membrane-associated VDR and activates signaling pathways such as PKA and PKC, following which, polyisoprenyl phosphate (PIPP) levels are elevated, thereby triggering the formation of inositol triphosphate (IP3). These signaling pathways help the entry of extracellular calcium into the cells or prompt the release of calcium from intracellular stores in the endoplasmic recticulum (ER). However, binding of 1α,25(OH)_2_D to the canonical nVDR causes a genomic response by initiating the transcription of targeted genes. Nemere et al. (2004), reported that 1,25-MARRS have a similar affinity for the ligand as that of the nVDR, but, the membrane-associated protein is 6–10 times more abundant in the cells than the nuclear receptor [107]. 1,25-MARRS is usually found associated with caveolin proteins. The intracellular Ca^2+^ levels are enhanced on binding of 1α,25(OH)_2_D to 1,25-MARRS. A study in keratinocytes showed that binding of 1α,25(OH)_2_D3 to the membrane receptor resulted in elevated metabolism of phosphatidylinositol (PI) to phosphatidylinositol triphosphate (PIP3), resulting in increased levels of IP3 in the cells [108,109]. The rise in IP3 were in accordance with a rise in calcium levels, eliciting a rapid response within 2–5 min [50]. Calcium is released from ER storage pools or through the transmembrane trafficking of calcium through the membrane calcium channels [110].

**Figure 6 ijms-20-03832-f006:**
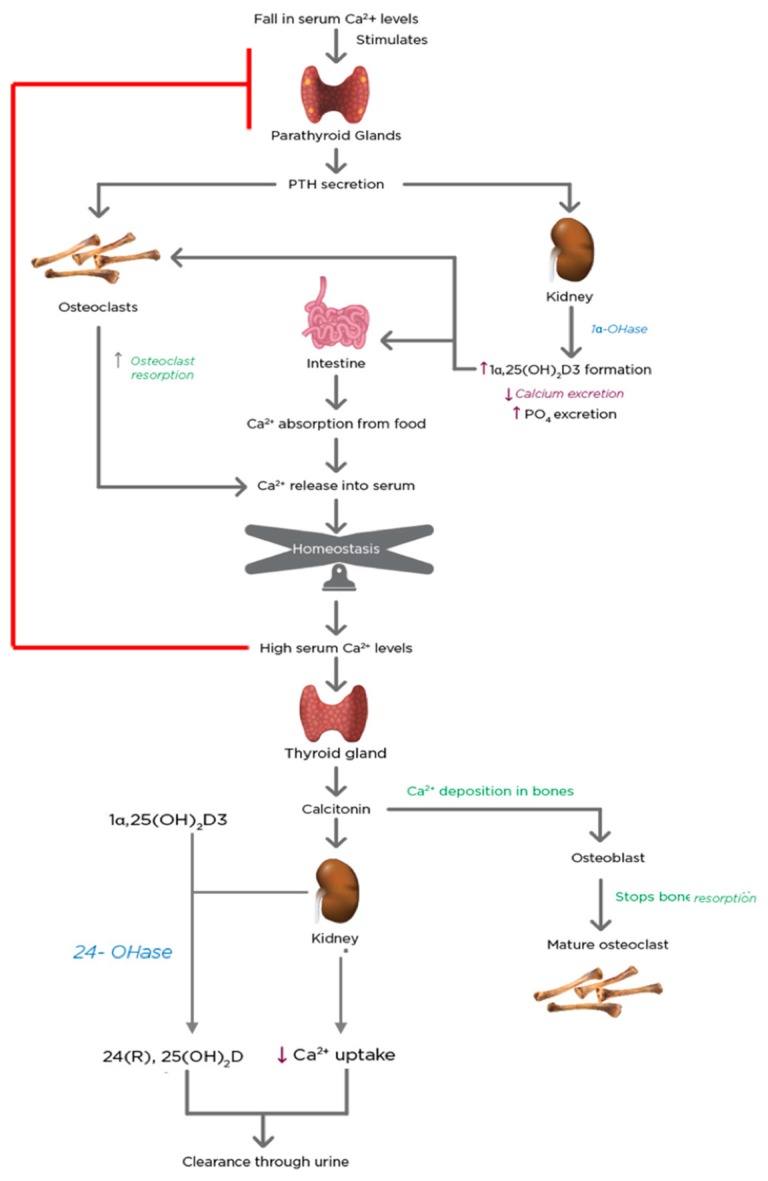
The regulation of mineral homeostasis by parathyroid hormone (PTH) and 1α,25(OH)_2_D3. The physiological functions of PTH and 1α,25(OH)_2_D3 are activated when serum calcium levels drop. The hormones act in conjunction with each other and exert coordinated effects on the kidneys, bones, and intestine to increase Ca^2+^ levels to normal. There is bone resorption, increased calcitriol formation by the kidneys and decreased calcium excretion from urine, and increased Ca^2+^ absorption by the intestine. Upon achievement of homeostasis, the process is shut down by a negative feedback loop, which is initiated by calcitonin secreted by the thyroid gland. Thus, the combined effect of PTH and 1α,25(OH)_2_D3 is necessary to maintain mineral homeostasis.

**Table 1 ijms-20-03832-t001:** The concentrations of circulating 25(OH)D in serum found in various biological conditions.

25(OH)D Levels	Condition
<30 ng/mL	Severe deficiency
30–50 ng/mL	Minor deficiency
50–70 ng/mL	Adequate levels
>80 ng/mL	Excess

**Table 2 ijms-20-03832-t002:** Direct target genes of 1α,25(OH)_2_D3, their location, and roles.

Target Gene	Cell Type	Function
*ASAP2 (ArfGAP with SH3 domain, ankyrin repeat and PH domain 2)*	Human THP-1 cells (monocytes)	Regulates autophagy, cellular migration and vesicular transport [44]
*CYP24A1 (Cytochrome P450 24A1)*	Kidney	Vitamin-D3 catabolizing enzyme [82]
*PTHLH (Parathyroid hormone-like hormone)*	Cytoplasm, golgi complex and nucleus of most cells	Activates PLC signaling pathways, proliferation of chondrocytes, regulation of bone formation by promoting recruitment and survival of osteoblasts, and plays a role in the physiological regulation of bone resorption [83]
*CAMP (Cathelicidin antimicrobial protein)*	Primary keratinocytes, monocytes, phagocytes, B cells and neutrophils	Antibacterial, antifungal and antiviral activities, the encoded protein functions in cell chemotaxis, immune mediator induction and inflammatory response regulation [84]
*Trpv6 (Transient Receptor Potential Vanilloid 6)*	Brush border membranes of the intestinal epithelia	Potential mediator of calcium uptake into the enterocyte [85]
*CYP27B1(Cytochrome P450 27B1)*	Kidneys, epithelial cells, lungs, breast, intestine, stomach, endocrine glands, cells of the immune system, osteoblasts chondrocytes	Expression of 1a-hydroxylase [44]
*CALB1*	Avian intestine and kidney and mammalian intestine, respectively	Codes for Calbindin-D28K and Calbindin-D9K proteins, upregulated by 1α,25(OH)_2_D3 [86,87]
*Osteocalcin*	Osteocytes, cartilages	Mineral deposition, bone resorption [86]
*FOXP3 (fork head box P3)*	Immune system	Maturity and performance of T regulatory cells [88]
*CD14 (Cluster of differentiation)*	Monocytes and most tissue macrophages and to a minor extent in monoblasts and promonocytes	Interacts with soluble lipopolysaccharide (LPS) released from gram-negative bacteria in combination with a plasma protein, LPS-binding protein [44]
*NINJ1 (Ninjurin 1)*	Hepatic stellate cells	Promotes axonal growth, may play a role in nerve regeneration and in the formation and function of other tissues [89]

**Table ijms-20-03832-t003a:** (**a**)

No.	References	Organism Studied	Form of vitamin D Used in the Study	Key Findings	Ref. No.
1	Sato K. (1993)	Rats	1α,25(OH)_2_D3	Elevated levels of 1α,25(OH)_2_D3 when administered to hypercalcemic patient with PTHrP-producing gastric carcinoma, aggravated the severity of malignancy-associated hypercalcemia (MAH).	[110]
2	Leigh-Clare J. (1927)	Australasian Petrel	Vitamin D	One of the first articles to describe the presence of Vitamin D in stomach oil. The study was an attempt to elucidate the source of vitamin D in the oil.	[136]
3	Selye H. and Bois (1957)	Sprague-Dawley rats	Ergocalciferol	VDT lead to calcium deposition in the muscularis of rat stomachs	[137]
4	Stumpf W. (1979)	Rats	1α,25(OH)_2_D3 and its metabolites	Vitamin D receptors for 1α,25(OH)_2_D3 or its metabolites target tissues of the GIT including the nuclei of some of the cells of the stomach	[138]
5	Kirui N. (1981)	Rats	1α,25(OH)_2_D3	1α,25(OH)_2_D3 has a direct role on hypercalcemia and STM with lesions forming in most of the soft tissues, including the gastric glandular mucosa and muscularis	[139]
6	Selking O. (1982)	Rats	Ergocalciferol	Vitamin D induced hypercalcemia in parathyroidectomized rats is associated with a thickened gastric mucosa, but the serum gastrin, number of gastrin (G) cells or antral gastrin remained unchanged	[140]
7	Kurose, T. (1988)	Rats	1α,25(OH)_2_D3	Calcium and 1α,25(OH)_2_D3 deficiency impair gastrin and somatostain secretion in the perfused rat stomach	[141]
8	Axelson J. (1991)	Rats	1α,25(OH)_2_D3	Gastrectomy may lead to an increase in the levels of serum 1α,25(OH)_2_D3, resulting in enhanced absorption of calcium by the small intestine. This may be the cause for diseases like osteomalacia in the gastrectomized patients.	[142]
9	Holick M. (1995)	Rats, mice, humans	1α,25(OH)_2_D3	Non-calcemic tissues including the gonads, pituitary gland, thymus, pancreas, stomach, breast, and skin possess the nuclear receptors for 1α,25(OH)_2_D3, targeting these receptors with analogs of 1α,25(OH)_2_D3 provides treatment against different diseases.	[143]
10	Stumpf W. (1995)	Two- month old mice	1α,25(OH)_2_D3 and its analogue 22-Oxacalcitriol (OCT)	Autoradiographic studies showed nuclear concentration and retention of 1α,25(OH)_2_D3 and its analogue 22-Oxacalcitriol (OCT) in neck mucus cells of gastric and pyloric glands and in dispersed endocrine cells in the antrum	[144]
11	Ikezaki S. (1996)	Male Wistar rats	24R,25(OH)2D3	The development of atypical hyperplasias and adenocarcinomas in the glandular stomachs was decreased by exposure to 24R,25(OH)2D3, which shows that 24R,25(OH)2D3 has chemopreventive effects.	[145]
12	Gagnemo- Persson R. (1999)	Male Sprague-Dawley rats	Ergocalciferol/Vitamin D2	Gastrin–ECL-cell axis can be suppressed by vitamin D or by vitamin D-dependent mechanisms. Also, vitamin D receptor gene expression was seen in the rat oxyntic mucosa.	[146]
13	Stumpf W. (2008)	Rats, mice, hamsters and zebra finch	1α,25(OH)_2_D3	Autoradiography studies confirmed the binding of 1α,25(OH)_2_D3 and its oxygen analog OCT in numerous regions of the digestive tract	[147]
14	Häkkinen I. and Lindgren I. (2009)	Albino rats	1α,25(OH)_2_D3	Excess of 1α,25(OH)_2_D3 leads to calcification of gastric tissues	[148]
15	Sahin H. (2018)	Rats	1α,25(OH)_2_D3	1α,25(OH)_2_D3 protects the gastric mucosa via attenuation of inflammatory reaction, oxidative stress and apoptosis.	[149]

**Table ijms-20-03832-t003b:** (**b**)

No.	First Author, Year	Form of Vitamin D Used in the Study	Key Findings	Ref. No.
1	Paterson C. (1965)	1α,25(OH)_2_D3	Gastrectomy and other surgeries of the stomach can trigger osteomalacia due to the increased levels of serum vitamin D which results in increased absorption of calcium by the small intestine.	[150]
2	Kimura K. (1967)	1α,25(OH)_2_D3	Calcium deposits were found in the soft tissues of a patient who died of uremia after administration of toxic levels of Vitamin D for 6 months	[151]
3	Lawrence, W. (1977)	1α,25(OH)_2_D3	Inadequate absorption of Vitamin D in patients who underwent gastrectomy is due to steatorrhea which may contribute to osteomalacia later on.	[152]
4	Giovannucci E. (2006)	1α,25(OH)_2_D3	Black men with poor vitamin D status are highly susceptible for digestive system cancer and have a higher mortality rate.	[153]
5	Chen W. (2007)	25(OH)D	A direct association between higher serum 25(OH)D concentration and increased risk of oesophageal squamous cell carcinomas (ESCC) in men but not women but no association with risk of gastric cardia or noncardia adenocarcinoma in either sex.	[154]
6	Pan L. (2010)	1α,25(OH)_2_D3	Studies in HGC-27 adenocarcinoma cells showed that Vitamin D can be used in gastric cancer therapies in association with trichostatin A ⁄ sodium butyrate and 5-aza-2-deoxycytidine	[155]
7	Antico A. (2012)	1,25(OH)2D	Low vitamin D concentration in *H. pylori* gastritis patients might lead to a more severe Th1-type aggression to the stomach epithelium	[156]
8	Park M.R. (2012)	Paricalcitol (19-nor-1,25-(OH)2D2)	Paricalcitol, has anticancer activity on GC cells by regulating cell cycle, apoptosis and inflammation	[157]
9	Kopic S. (2013)	1α,25(OH)_2_D3	Gastric acid, gastrin secretion is increased in humans under conditions of hyperparathyroidism with the combined actions of both PTH and vitamin D	[158]
10	Guo L. (2014)	1α,25(OH)_2_D3	VDR and CAMP expression is up-regulated in the gastric epithelium during *H. pylori* infection; thus, VDR is important in maintaining the homeostasis of gastric mucosa and protection from *H. pylori* infection.	[159]
11	Wen Y (2015)	1,25(OH)_2_D3	VDR expression was seen to decline from the premalignant stage, to low expression in gastric cancer tissues. VDR could be a potential prognostic factor for patients with gastric cancer.	[160]
12	Bashir M. (2016)	1α,25(OH)_2_D3	Vitamin D influences the composition of gut microbiome.	[161]
13	Vyas N. (2016)	25(OH)D	There is a positive relationship between VDD and gastric adenocarcinoma	[162]
14	Du C. (2017)	1α,25(OH)_2_D3	Vitamin D supplement might be a safe and economical way to prevent or treat gastric cancer	[163]
15	Yildirim O. (2017)	25(OH)D	Deficiency of 25(OH)D may be a risk factor related to eradication failure of *H. pylori*, and supplementation of 25(OH)D before eradication of *H. pylori* may provide better results.	[164]
16	El Shahawy M.S. (2018)	25(OH)D3	25(OH)D deficiency can be a risk factor in eradication failure of *H. pylori* infection.	[165]
17	Fletcher J. (2019)	1α,25(OH)_2_D	1α,25(OH)_2_D protects the gastrointestinal barrier and evokes immune responses	[166]

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
