# Peer review of "A Narrative Role of Vitamin D and Its Receptor: With Current Evidence on the Gastric Tissues"

_ijms, 2019, doi:10.3390/ijms20153832_

Round 1

Reviewer 1 Report

This is a well-written and comprehensive review which addresses multiple aspects of vitamin D chemistry and biology as well as of VDR functions and roles in the regulation of tissue homeostasis in health and disease, focusing on the gastric tissue. Still, the manuscript would benefit from clarifying and/or elaborating on a few points below:

Line 11: “Vitamin D or 1a,25(OH)2D is a major steroid hormone that is gaining attention as both a nutrient and a therapeutic molecule… ” – It is not so accurate to directly identify vitamin D with 1α,25(OH)2D. For instance, 1α,25(OH)2D is not “a nutrient”. 

Line 52: “The non-classical functions of vitamin D and its importance in human physiology became evident ONLY OVER THE PAST DECADE [18].” – Several non-classical functions of vitamin D compounds became evident much earlier, in the early 80’s (e.g. antiproliferative and differentiation-inducing activities as well as effects on the immune system). The authors need to revise the phrase and, probably, the reference. In this respect, the following phrase that appears in Section 6 (line 483) seems to be more realistic: “Notably, in the last FOUR DECADES, non-calcemic effects of 1α,25(OH)2D have been explored and investigated”.  

Line 500: “Tables 3 (a) and (b) reference the MAJOR available studies that emphasize the key findings in the animal and human stomach.” – What criteria did the authors use to select “major available studies”?  What databases were searched for the analysis of the available relevant literature? A simple search strategy using the keywords “Vitamin D in Stomach” (line 503) does not seem to be sufficient for such analysis. For example, the use of a more complex search string “(vitamin D OR calcitriol OR dihydroxyvitamin) AND (stomach OR gastric)” and only one search engine - PubMed, which mainly accesses the MEDLINE database, - produces ~740 relevant hits, of which ~150 are reviews. The authors may increase the value of the “Vitamin D and the gastric tissue” section by implementing a more extensive search strategy for the analysis of the related publications.

Author Response

Dear Reviewer

Thanks for your efforts. Attached the responses to your comments.

Best Regards

Asma

Reviewer 2 Report

The authors includes more detail about vitamin D, vitamin D status and its receptor.

However, the title may be change because the information about the gastric field is a little part of the manuscript.

In conclusion may be described the importance of vitamin D and its mechanism on gastric tissue in order to improve the information on human health.

Author Response

(The authors gave the same response as above.)

Reviewer 3 Report

In this review authors summarized the central role of vitamin D in gastric and in preserving and maintaining the overall the health and homeostasis in gastric tissues. The paper also provides the basic knowledge about different forms of vitamin D, vitamin D receptor and signaling transduction pathways that regulated by vitamin D. This review is well organized and written about the important information in the field.  However, below suggested changes need to be addressed for a better understanding of the readers.

1.        It would be nice to list the abbreviations that used in the review so that it will be easier to follow. 

2.       In Figure 2, “Pre-vitamin D” should be “Pre-vitamin D3”.

3.       25(OH)D was used in the beginning, then 25OHD later in the review, please keep it consistent through the whole paper, it also apply for other key words.

4.       Vitamin D2 and D3 are totally different, so try not use general tag of 1,25(OH)2D for both of them.

5.       In table 3, you may also need to search with different key words, such as “vitamin D in gastric tissue”, so you will not miss any important papers.

6.       In Figure 6, there is an extra rectangle cover the description in the bottom of the chart.

Author Response

(The authors gave the same response as above.)
